# Batch Differentiable Pose Refinement for In-The-Wild Camera/LiDAR Extrinsic Calibration

**Lanke Frank Tarimo Fu**
University of Oxford
fu@robots.ox.ac.uk

**Maurice Fallon**
University of Oxford
mfallon@robots.ox.ac.uk

Misaligned ⟶ Aligned

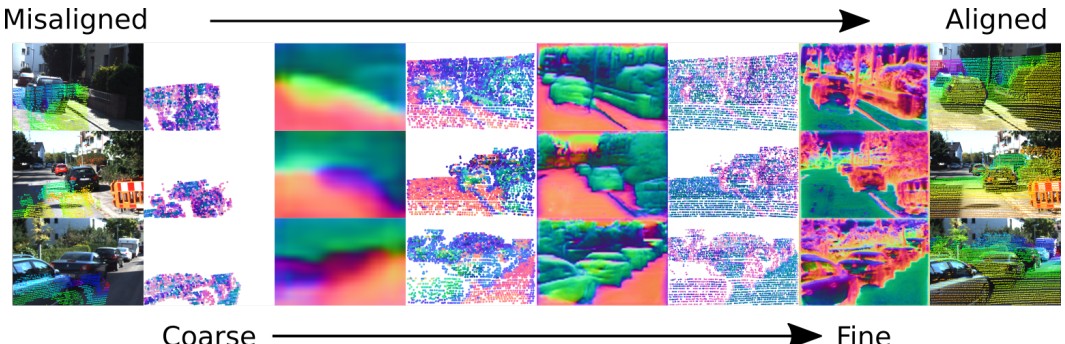

Coarse ⟶ Fine

Figure 1: Coarse-to-fine refinement of the LiDAR-to-camera extrinsic parameters. The matching appearance between the LiDAR and camera features is trained using only ground-truth extrinsic parameters for self-supervision. During training, batched refinement helps retain difficult samples that individually would have been discarded. During inference, we show that batched refinement achieves state-of-the-art zero-shot transfer. The rightmost column shows the refined overlay of LiDAR points in the image.

**Abstract:** Accurate camera to LiDAR (Light Detection and Ranging) extrinsic calibration is important for robotic tasks carrying out tight sensor fusion — such as target tracking and odometry. Calibration is typically performed before deployment in controlled conditions using calibration targets, however, this limits scalability and subsequent recalibration. We propose a novel approach for target-free camera-LiDAR calibration using end-to-end direct alignment which doesn't need calibration targets. Our batched formulation enhances sample efficiency during training and robustness at inference time. We present experimental results, on publicly available real-world data, demonstrating 1.6cm/0.07° median accuracy when transferred to unseen sensors from held-out data sequences. We also show state-of-the-art zero-shot transfer to unseen cameras, LiDARs, and environments.

**Keywords:** Sensor Fusion, Extrinsic Calibration, Differentiable Optimization

## 1 Introduction

In many multi-sensor robotic setups, information fusion between any two sensors requires accurate knowledge of the relative transformation between the sensors — the extrinsic parameters. In the case of sensor fusion between a camera and a LiDAR, the extrinsic parameters along with the intrinsic parameters of the camera are used to determine point-to-pixel correspondence displayed in Fig. 1. This correspondence enables fusion in downstream tasks such as object detection, tracking, and ego-motion estimation. Camera/LiDAR extrinsic calibration 'in-the-wild' — meaning in uncontrolled environments without specialized targets — is difficult due to the domain gap between the two sensors. Cameras register textural information but can't directly measure geometry whereas LiDARs

7th Conference on Robot Learning (CoRL 2023), Atlanta, USA.

measure geometry but can't register texture. Additionally, cameras are passive sensors and may capture illumination variations like shadows. In contrast, LiDARs, which produce their own light, don't detect shadows in the same visual range. Given these challenges, in-the-wild camera/LiDAR extrinsic calibration is still an active research question.

In this work, we present a framework for camera/LiDAR extrinsic calibration that:

- Can recover the accurate extrinsic parameters from a wide range of initial estimates.

- Learns relevant features for pose alignment from both camera and LiDAR input automatically.

- Generalizes to sensors and environments not encountered during training.

We formulate the camera/LiDAR extrinsic calibration problem as one of batch differentiable direct alignment – aligning a batch of learned features from the LiDAR domain to their corresponding batch of deep image features, guided by deep feature gradients in the image. The end-to-end differentiable nature of our formulation relaxes the need for manual feature tuning. We show in our experiments that not only do we achieve state-of-the-art performance on the training sensor suite (with median translation and rotation errors of 1.6cm and 0.07°), but our method also generalizes to unseen sensor models and environments — with results demonstrated using a variety of common datasets.

## 2 Related Works

Classical solutions to the problem of target-free camera/LiDAR extrinsic calibration form two categories: The first category exploits correlations between the image intensity value and the LiDAR reflectance value [1, 2, 3]. These methods perform well under assumptions of uniform lighting in the environment but still require an initial guess which is close to the ground truth extrinsic parameters. The other category of methods maps geometric (e.g. depth or normal) discontinuities in the LiDAR scan to image intensity discontinuities i.e. image gradients [4, 5, 6]. This use of local image gradient information can be augmented by local intensity normalization, which improves the robustness of these methods in scenes with varying illumination sources. Still, their performance in cluttered environments is limited by the abundance of local minima. Both these categories of classical methods typically require manual parameter tuning for each unique sensor suite and as such struggle to be adapted to different configurations and manufacturers.

More recent works tackle the target-free calibration problem using deep-learned approaches. We classify these works into two categories: i) regression-based methods [7, 8] that align camera and LiDAR features using parameters regressed by a deep neural network. While these methods demonstrate impressive accuracy on held-out datasets of their training environment, their regression-based nature makes a zero-shot transfer to other datasets difficult since the neural network is biased to output extrinsic parameters in the distribution seen during training. Another, newer category of methods employs differentiable optimization for either (ii) indirect feature point matching via non-linear perspective-n-point [9]; or (iii) calibration-flow refinement [10] to align LiDAR-extracted deep features to their corresponding pixel locations in the image. While both these methods generalize better than regression-based methods, [10] only recovers calibration over a narrow region of operation and [9] achieves significantly less accurate results.

RGKCNet [9] and DXQNet [10] are of a different category of differentiable pose alignment compared to our method, in that they are forms of indirect alignment. RGKCNet learns 3-D point to 2-D pixel correspondences, and thus incorrectly models degenerate features such as lamp posts and thin trees which are line constraints and not point constraints. DXQNet learns a 2-dimensional weighting term which is used to weight the per-point alignment along XY-axis aligned directions. This puts the burden on the network to not just learn features but to also accurately model their orientations. Our method overcomes this challenge by explicitly computing deep image gradients, naturally dis-

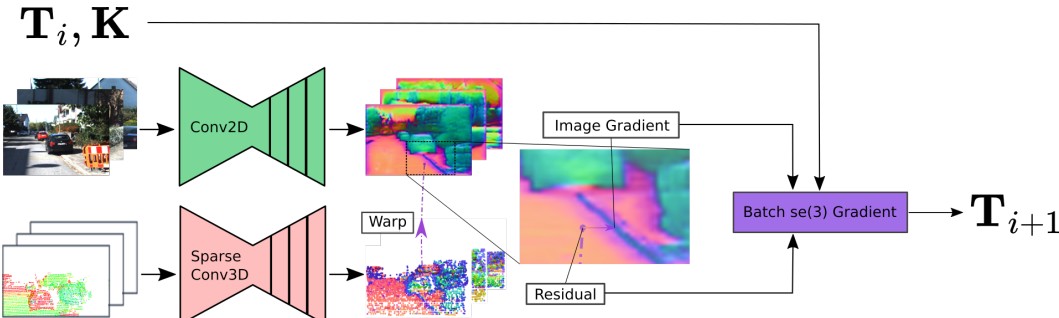

Figure 2: An overview of one iteration of the batched differentiable alignment. Batches of pairs of images and voxelized point clouds are passed through their respective U-Net feature extractors. Each sparse voxel feature is warped to the image using the latest transform parameters and the camera intrinsic parameters and registers a residual for its difference to the image feature at the corresponding pixel location. This residual along with the image gradient and the Jacobian of the projected pixel location with respect to the transform parameters forms the signal for the optimization.

tinguishing between line and point constraints — allowing the network to focus simply on learning useful features.

## 3    Method

We take inspiration from methods for scene-agnostic visual localization. Like PixLoc [11], we avoid overfitting pose estimation into a scene memorization problem by performing deep feature alignment using differentiable optimization. This allows us to decouple the task of projective geometry, in pose refinement, from the task of learning features. In doing so, we translate the task of achieving zero-shot transfer for camera/LiDAR extrinsic calibration, into the task of learning camera and LiDAR features that transfer to unseen environments, captured by different sensors.

### 3.1    Problem formulation

Given a point cloud, a set of $N$ 3-D points, $\mathbf{P}_L \in \mathbb{R}^{3 \times N}$ represented in the LIDAR reference frame (denoted by the subscript $L$) and a set of images, we want to determine the extrinsic parameters $\mathbf{R}_L^C \in \mathbf{SO}(3), \mathbf{t}_L^C \in \mathbb{R}^3$, such that,

$$\mathbf{P}_C = \mathbf{R}_L^C \mathbf{P}_L + \mathbf{t}_L^C. \tag{1}$$

Where $\mathbf{P}_C$ are the corresponding coordinates of $\mathbf{P}_L$, from the camera's reference frame. For brevity, we use $(\mathbf{R}_L^C, \mathbf{t}_L^C) = \mathbf{T}_L^C \in \mathbf{SE}(3)$ to denote the rigid body transform and omit the reference frame scripts i.e. $\mathbf{T} \triangleq \mathbf{T}_L^C$.

### 3.2    Camera/LiDAR extrinsic calibration as differentiable direct alignment

With direct alignment, the transform parameters are optimized to minimize the misalignment in the appearance of the signals registered by different sensors. In our case, our two sensors are the LiDAR and the camera. Instead of attempting to match the raw outputs from sensors, such as the intensity from LiDAR to image intensity, which has proven problematic in difficult lighting situations [3], we begin by extracting deep features from the raw inputs.

From the LiDAR side, we map the 3-D points using the initial guess of the extrinsic parameters to get $\hat{\mathbf{P}} = \hat{\mathbf{R}} \mathbf{P}_L + \hat{\mathbf{t}}$. We then voxelize these points and extract, using a sparse 3-D CNN, a set $(\hat{\mathbf{P}}^p, \mathbf{F}_L^p, \mathbf{w}_L^p)$ for each $p \in [1, ..., P]$ of a $P$-level multi-scale feature pyramid. At each level, $\hat{\mathbf{P}}^p \in \mathbb{R}^{3 \times N_p}$ are the $N_p$ voxel centroids, and $\mathbf{F}_L^p \in \mathbb{R}^{D_p \times N_p}$ are the $N_p$ corresponding deep feature vectors at each voxel centroid of dimension $D_p$. Lastly, $\mathbf{w}_L^p$ is the vector of $N_p$ learned weights with elements in the range $[0, 1]$.

From the camera side, we use a 2-D CNN to extract from the camera image $\mathbf{I} \in \mathbb{R}^{W \times H \times 3}$ a pyramid of deep features $\mathbf{F}_C^p \in \mathbb{R}^{W_p \times H_p \times D_p}$, and corresponding weights $\mathbf{W}_C^p \in \mathbb{R}^{W_p \times H_p}$. Similarly, $p \in [1, ..., P]$ stands for one of the pyramid levels, each matching in scale with their corresponding LiDAR extractor pyramid levels.

At each $i$-th iteration of the optimization, the misalignment between the deep features extracted from the $j$-th LiDAR point and camera features at its corresponding projected point is given by,

$$\mathbf{r}_j^p = (\mathbf{F}_{L_j}^p - \mathbf{F}_C^p[\Pi^p(\mathbf{R}_i \hat{\mathbf{P}}_j^p + \mathbf{t}_i)]) \in \mathbb{R}^{D_p}. \tag{2}$$

$\Pi^p$ is the projection function of a 3-D point in the camera reference frame onto the image plane at level $p$, and $[\cdot]$ denotes sub-pixel lookup by bi-linear interpolation. We've also introduced new variables $\mathbf{R}_i$ and $\mathbf{t}_i$ which are optimization parameters that we iterate and initialize as $\mathbf{R}_0 = \mathbf{I}^{3\times3}$ and $\mathbf{t}_0 = \mathbf{0}$.

Similar to [11], we formulate a total cost function from these residuals in the form,

$$E_p(\mathbf{R}_i, \mathbf{t}_i) = \sum_j^{N_p} w_{l_j} w_{c_j} \rho(\|\mathbf{r}_j^p\|^2), \tag{3}$$

where $w_{l_j}$ is the $j$-th element of $\mathbf{w}_L^p$, the LiDAR-learned weighting, and $w_{c_j}$ is the camera-learned weighting sub-pixel interpolated at the pixel location of the $j$-th LiDAR point projected into the image plane at level $p$. Lastly, $\rho$ is the learnable robust cost function [12]. To robustly minimize this non-linear least-squares problem we perform the alignment in a coarse-to-fine fashion, using solutions from the previous coarser level of the pyramid as the starting point in the problem of the finer level. At each level, we use the learned Levenberg-Marquardt algorithm [11], parameterizing transform updates using $\xi \in \mathfrak{se}(3)$. As such, we stack all the residual terms at level $p$ into $\mathbf{r}^p \in \mathbb{R}^{D_p N_p}$ and formulate each row of the Jacobian $\mathbf{J} \in \mathbb{R}^{D_p N_p \times 6}$ and the Hessian $\mathbf{H} \in \mathbb{R}^{6\times6}$ with respect to $\xi$ as,

$$\mathbf{J}_{k+(j-1)D_p} = \frac{\partial \mathbf{r}_{j_k}^p}{\partial \hat{\mathbf{p}}} \frac{\partial \hat{\mathbf{p}}}{\partial \xi}, \text{ and } \mathbf{H} = \mathbf{J}^\mathsf{T} \mathbf{W} \mathbf{J}, \tag{4}$$

where $j \in [1, N_p]$ is the $j$-th 3-D point, $k \in [1, D_p]$ denotes the $k$-th dimension of the deep-learned features at level $p$, and $\mathbf{W} \in \mathbb{R}^{D_p N_p \times D_p N_p}$ is a block diagonal matrix with $N_p$ blocks of size $D_p \times D_p$ each, where the $j$-th block has uniform diagonal weights $w_{l_j} w_{c_j}$. With these defined, we compute a Gauss-Newton gradient step in the direction of decreasing cost with $\xi = -\mathbf{H}^{-1}\mathbf{J}^\mathsf{T}\mathbf{W}\mathbf{r}$, and update our transform estimate using,

$$\begin{bmatrix} \mathbf{R}_{i+1} & \mathbf{t}_{i+1} \\ \mathbf{0} & 1 \end{bmatrix} = \exp(\widehat{\xi})^\mathsf{T} \begin{bmatrix} \mathbf{R}_i & \mathbf{t}_i \\ \mathbf{0} & 1 \end{bmatrix}. \tag{5}$$

We visualize one of these steps in Fig. 2

**The training loss:** During training, we perform these alignment updates for fixed $M$ steps at each pyramid level, yielding for each level, $\mathbf{T}_M^p$. We supervise these transforms with the ground truth using the reprojection error of each 3-D point:

$$\mathcal{L} = \sum_p \sum_j \rho(\|\Pi^p(\bar{\mathbf{R}}\hat{\mathbf{P}}_j^p + \bar{\mathbf{t}}) - \Pi^p(\mathbf{R}_M^p \hat{\mathbf{P}}_j^p + \mathbf{t}_M^p)\|^2). \tag{6}$$

Note that since the points $\hat{\mathbf{P}}_j^p$ are LiDAR points projected by the initial guess of the extrinsic parameters, our supervision signal $\bar{\mathbf{T}}$ is given by, $\bar{\mathbf{T}} = \mathbf{T}\hat{\mathbf{T}}^{-1}$, where $\mathbf{T}$ is the ground truth extrinsic parameters and $\hat{\mathbf{T}}$ is the initial guess applied to the LiDAR points before voxelization.

### 3.3 Batch SE(3) alignment

**At test time:** When we make the added assumption that we are solving for the same transform parameters across a batch of image/point cloud pairs at test time, the change to our algorithm is

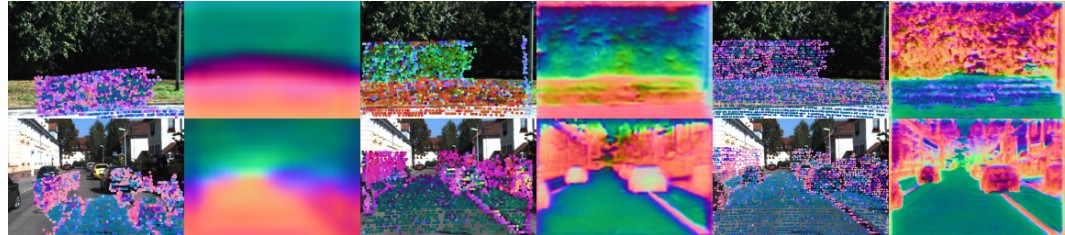

Figure 3: PCA feature visualization of a feature-deprived scene (top) and a feature-rich scene (bottom). During training, our batch formulation keeps the harder example at the top from diverging. Consequently, we can still learn the sparse amount of features it does have e.g. the ground to hedge differences in the middle columns, and the salient post on the rightmost column.

rather straightforward. Where we used to have, at each pyramid level, the residual $\mathbf{r}^p \in \mathbb{R}^{D_p N_P}$, we now stack all the residuals across the batch together and get the vector $\mathbf{r}_B^p \in \mathbb{R}^{D_p N_B}$ where $N_B = \sum_b^B N_b^p$ is the total number of 3-D points in the batch at level $p$. Then, we update Eq. (4) accordingly and solve the optimization steps just as we did in the single sample case. This technique is deployed in most existing non-learning-based target-free calibration works [4, 3, 2], where it is interpreted as making the calibration cost function Eq. (3) smoother and more convex. To our surprise, this simple yet very effective scheme has so far not been deployed in deep-learned camera-LiDAR extrinsic calibration.

**At training time** While at first glance it may seem impossible to perform batch pose alignment during training since we might encounter camera/LiDAR pairs with heterogeneous extrinsic parameters, note that the pose alignment is performed relative to the initial guess (Eq. (5)). So while the actual camera/LiDAR relative transforms may differ across the batch, we can independently pick initial guesses for each sample in the batch such that the relative transform from the initial guess to the ground truth is identical. Mathematically, this gives for each $b$-th sample in the batch:

$$\hat{\mathbf{T}}_b = \Delta \mathbf{T}^{-1} \mathbf{T}_b. \tag{7}$$

Initializing the initial guess of each sample using Eq. (7) enables joint optimization across all samples of the entire batch, allowing us to learn features from even hard examples where individually the optimization would have diverged (see Fig. 3).

## 4 Training Setup

To test the capability of our framework to perform accurate zero-shot transfer to unseen environments, we've set up our experiments to train solely on one dataset, and subsequently evaluate performances using other datasets with different sensors and test environments.

**Dataset:** Due to its popularity as a benchmark for learning-based camera/LiDAR calibration, we use the KITTI Odometry dataset [13] as our sole training dataset. It consists of 22 sequences of sub-urban driving scenarios. During training, we only use camera "2" which is a front-facing colored perspective camera and the top-mounted Velodyne HDL-64E LiDAR. Of the 22 sequences, we use sequences "01" – "21", leaving sequence "00" out for validation and testing.

**Setup for LiDAR and camera input:** Different LiDARs exhibit different spatial coverage, intensity profiles, and reference coordinate frames. To make our framework robust to these variations during zero-shot transfer, we perform pose and intensity augmentations to the LiDAR point cloud and crop augmentations to the camera image. We specify these details in Appendix A.1.

**Model:** To facilitate robust calibration from large initial offsets, we use a coarse-to-fine alignment scheme with 3 levels. Two U-Net [14] architectures extract deep features from LiDAR and camera separately for each of these levels. To aid the correspondence of similar features from different domains, we further adapt the features from each domain with a single multi-layer perceptron. We provide details about our model and its weight initialization in Appendix A.2. During training, we

Table 1: Results on the **same camera** of a held-out sequence (values show component-wise mean/median absolute values of translation/rotation along each axis).

| Initial error | Method | Mean/Median $\Delta$t (cm) | | | Mean/Median $\Delta$R (°) | | |
|---|---|---|---|---|---|---|---|
| | | x | y | z | roll | pitch | yaw |
| $\pm$1.5m $\pm$20° | LCCNet | **0.24/0.26** | **0.38/0.36** | **0.46/0.35** | **0.03**/0.03 | **0.01/0.00** | **0.04/0.02** |
| | DXQNet | / | / | / | / | / | / |
| | Ours (1) | 8.77/1.76 | 5.50/1.45 | 9.25/1.80 | 0.36/0.08 | 0.43/0.07 | 0.53/0.07 |
| | Ours (8) | 2.26/0.51 | 2.02/0.87 | 1.24/0.58 | 0.10/**0.02** | 0.21/0.04 | 0.16/0.03 |
| $\pm$0.1m $\pm$5° | LCCNet | 0.24/0.15 | **0.48/0.26** | 1.11/0.47 | **0.02/0.02** | 0.17/0.10 | 0.03/0.03 |
| | DXQNet | 0.75/0.53 | **0.48**/0.51 | 1.09/0.78 | 0.05/0.03 | 0.05/**0.03** | 0.03/**0.02** |
| | Ours (1) | 3.23/0.94 | 2.58/1.04 | 3.42/1.18 | 0.09/0.05 | 0.13/0.05 | 0.15/0.04 |
| | Ours (8) | **0.42/0.32** | 0.82/0.83 | **0.59/0.46** | **0.02/0.02** | **0.04**/0.04 | **0.02/0.02** |

use the Adam [15] optimizer with a learning rate of $10^{-5}$ to train for 20 epochs with a batch size of 8.

# 5    Results

All results showcased here are derived from models trained on a single camera/LiDAR pair as detailed in Section 4. To assess the capacity for zero-shot transfer, we incrementally test on more challenging scenarios, starting with a held-out sequence of the training data and culminating in tests on completely different cameras, LiDARs, and environments.

## 5.1    Extrinsic calibration in settings similar to training

**Testing using the same camera:** In this simple setting, we use images from the training camera (camera "02") but from a held-out sequence "00" of the KITTI Odometry dataset which comprises 4541 image/point cloud samples. The differences between the extrinsic parameters in this sequence and the training sequences are negligible, so the key distinction from the training data is the novel scenes in this held-out sequence.

We compare against the regression-based LCCNet [8] and the differentiable calibration flow method DXQNet [10]. When comparing against LCCNet, we pass to our model initial extrinsic parameter guesses sampled uniformly [0, 20]°and [0, 1.5]m around the ground truth value, using the scheme presented in Appendix A.1. Note that the initial angular errors used in this experiment are even larger than the values we used during training.

The upper section of Table 1 shows LCCNet outperforming our method. Our method performs poorly on single image/point cloud pairs due to scenes with insufficient data for full 6-DoF pose observability. However, our method substantially improves when run in batch optimization using 8 pairs, achieving sub-centimeter median absolute error on each translation axis and even surpassing LCCNet in median roll rotation accuracy. Remarkably, our model achieves this despite the fact that it has never encountered rotation perturbations up to 20°during training.

DXQNet is designed to only recover calibration from small drifts [10] in the range [0, 5]°and [0, 0.1]m, so we sample initial calibration parameters in this same range when comparing our method against DXQNet. The lower section of Table 1 shows that our method's single-sample performance is slightly worse than DXQNet in median metrics and notably worse in mean metrics since our method diverges when a single scene lacks sufficient structure. However, performing batch optimization significantly enhances our method, achieving lower error than DXQNet along all axes except for Y-axis translation and median pitch-angle error.

While DXQNet is designed only for calibration from small initial errors of [0, 5]°and [0, 0.1]m [10], as seen in Table 1, our method competes even against state-of-the-art regression-based methods like LCCNet and recovers calibration from large initial errors of [0, 20]°and [0, 1.5]m.

Table 2: The performance change: trained on camera "2" and tested on camera "3" of a held-out sequence (values show the mean/median magnitudes of the translation/rotation vector).

| Initial error | Method | Mean $\Delta$t (cm) | | Mean $\Delta$R (°) | | Median $\Delta$t (cm) | | Median $\Delta$R (°) | |
|---|---|---|---|---|---|---|---|---|---|
| | | seen | unseen | seen | unseen | seen | unseen | seen | unseen |
| $\pm$1.5m $\pm$20° | LCCNet | **1.59** | 52.5 | 0.16 | 1.54 | **1.01** | 52.5 | 0.12 | 1.47 |
| | DXQNet | / | / | / | / | / | / | / | / |
| | Ours (1) | 15.98 | 20.15 | 0.92 | 1.07 | 3.60 | 4.65 | 0.16 | 0.20 |
| | Ours (8) | 3.09 | **3.77** | **0.15** | **0.30** | 1.39 | **1.69** | **0.07** | **0.07** |
| $\pm$0.1m $\pm$5° | LCCNet | 1.29 | 52.5 | 0.18 | 1.52 | **0.61** | 52.5 | 0.12 | 1.47 |
| | DXQNet | 1.43 | 2.94 | 0.08 | 0.16 | 0.81 | 2.28 | 0.07 | 0.13 |
| | Ours (1) | 6.25 | 8.32 | 0.25 | 0.31 | 2.21 | 2.86 | 0.10 | 0.13 |
| | Ours (8) | **1.20** | **1.76** | **0.05** | **0.07** | 1.12 | **1.65** | **0.06** | **0.07** |

**Testing using a different camera at a different vantage point:** While all the methods are trained on camera "2" (seen), in this test, we perform calibration between the LiDAR and camera "3" (unseen). This is significant for generalization because cameras "3" and "2" are separated 50 cm apart. We also use two ranges of initial errors in this experiment, a larger one to compare against LCCNet, and a smaller one for DXQNet. In doing so, we test our model's ability to both recover calibration from large initial errors and also transfer to new sensors.

Unlike the case of testing on the *seen* camera, the upper section of Table 2 shows that, when tested on the *unseen* camera, our method consistently performs better than LCCNet — whose mean and median translation error magnitudes (of more than 50cm) are more than a magnitude higher than the 1cm error achieved with the *seen* camera. While our method also experiences a drop in accuracy when tested on the *unseen* camera, our drop is significantly lower (from 3.60cm to 4.65cm median translation error), and even lower when run in batch optimization, from 1.39cm to 1.69cm median translation error magnitude – only a 3mm drop.

Meanwhile, DXQNet, the learned calibration flow method, is more robust when transferred to the *unseen* camera. Seen on the lower section of Table 2, the median translation accuracy of DXQNet drops to 2.28cm on the *unseen* camera, which is still relatively accurate when compared to LCCNet. While DXQNet performed better than our method in the single image/point cloud pair setting on the *seen* camera, once transferred to the *unseen* camera, the performance gap narrows down. In fact, shown on the lower section of Table 2, our method and DXQNet achieve the same level of median rotation accuracy.

We see that running batched optimization with our method achieves the best mean/median rotation and translation accuracy compared to all other methods when transferred to the *unseen* camera. Additionally, the drop in accuracy (transferring from the *seen* to the *unseen* camera) is lower using our method – especially so in the batched alignment case. We highlight these facts in Fig. 4, where the slope of the graphs highlights the drop in accuracy.

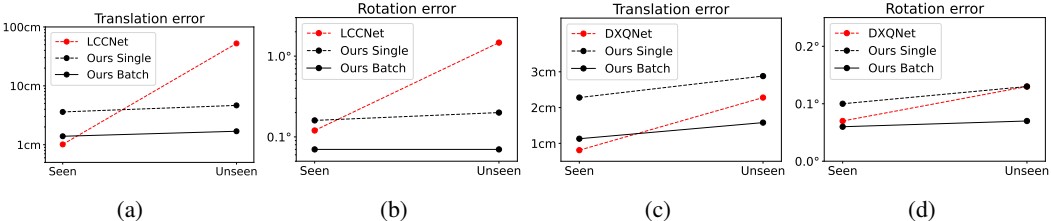

(a)  (b)  (c)  (d)

Figure 4: Slope graphs highlighting the error increase when the methods are tested on an unseen camera, having trained on the seen camera. All plots show median metrics: (a) and (b) comparing translation and rotation errors of our method versus LCCNet, and (c) and (d) comparing against DXQNet. Note that our method exhibits gentler slopes compared to the other methods, showing a more robust transfer to the unseen camera.

Table 3: Zero-shot transfer performance on different datasets.

| Method | $\Delta t$ (cm) | $\Delta R$ (°) | Method | $\Delta t$ (cm) | $\Delta R$ (°) |
|---|---|---|---|---|---|
| DXQNet | 5.65/4.70 | 2.89/1.03 | LCCNet | 324/318 | 20.8/18.1 |
| Ours Single | 28.0/5.99 | 1.70/0.55 | Ours Single | 102/16.8 | 3.60/0.64 |
| Ours Batch | **3.67/3.61** | **0.51/0.51** | Ours Batch | **6.97/3.87** | **0.44/0.43** |

(a) KITTI-360 dataset. Initial error in the range: $\pm 5°\pm 0.1$m. Ours Batch uses a batch size of 8

(b) Waymo dataset. Initial error in the range: $\pm 20°\pm 1.5$m. Ours Batch uses a batch size of 4

## 5.2 Zero-shot transfer to different environments with different sensors

**Transfer to a different camera in a different environment:** The KITTI-360 [16] dataset is captured in Karlruhe, just like the KITTI Odometry dataset we used during training. However, the camera setup is different both in intrinsic parameters and its relative transform to the LiDAR.

We compare our calibration accuracy to DXQNet, as they have also reported their zero-shot transfer metrics in the KITTI-360 setup. In Table 3a, we show that overall, the accuracy achieved by both methods is worse in rotation and in translation compared to their respective performances on the KITTI Odometry dataset. While our single sample optimization method performs better in rotation but worse in translation than DXQNet, our batched optimization method (batch of 8) performs significantly better than DXQNet in both rotation and translation.

**Transfer to a different camera, LiDAR, and environment:** We also test generalizability to the Waymo dataset [17], which has a higher resolution camera and a custom LiDAR. To match the training data image resolution, we halve the image size and camera intrinsics, allowing for consistent feature extraction. This flexibility further distinguishes optimization-based calibration methods from regression-based methods which can't explicitly reconfigure the camera projection parameters.

In Table 3b, we see that LCCNet does not generalize, with translation errors over 3m and rotation errors at 20.8°(mean) and 18.1°(median). Our method, run on a single sample, is poor in the mean metric (just above 1m and 3.6°), but is better in the median metric (16.8cm, 0.64°), suggesting that some outlier scenes impact calibration performance. Notably, batch optimizing image/point cloud pairs improves our performance significantly, reducing the median translation error to 3.87cm and rotation errors to 0.44°(mean) and 0.43°(median).

## 6 Limitations

Currently, our model assumes shared visibility between the LiDAR and the camera, enabling pose alignment from simultaneous image and point cloud pairs. In sensor settings without shared visibility, existing literature resolves this by creating a local map from several images and LiDAR scans [18]. Our model further presumes simultaneous image pixel and LiDAR point registration, necessitating ego-motion compensation for rotating LiDAR models. To overcome these limitations, in future work, we aim to tackle ego-motion estimation, inter-sensor temporal calibration and LiDAR/camera extrinsic calibration jointly using differentiable representations of sensor relative pose as explored in [19].

## 7 Conclusion

We have presented a method for in-the-wild camera/LiDAR calibration that both recovers calibration from large initial errors ([0, 20]°and [0, 1.5]m) and transfers to unseen sensors and environments — a trait that no existing method has demonstrated. While these are promising results for in-the-wild calibration, our accuracy still falls short of target-based calibration methods. In future work, we aim to incorporate more geometric priors such as mapping/ego-motion consistency into our feature learning to facilitate online tasks that require higher degrees of accuracy.

## Acknowledgments

Support for this work has been provided by the Horizon Europe project DigiForest (101070405) and a Royal Society University Research Fellowship (M. Fallon). This work has been carried out within the framework of the EUROfusion Consortium, funded by the European Union via the Euratom Research and Training Programme (Grant Agreement No 101052200 — EUROfusion). Views and opinions expressed are however those of the author(s) only and do not necessarily reflect those of the European Union or the European Commission. Neither the European Union nor the European Commission can be held responsible for them.

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

# A   Details on experimental setup

## A.1   Setup for LiDAR and camera input

**Setup for LiDAR input:** To learn robust features for the calibration of different transforms, we train our network to recover the true calibration from a variety of initial guesses $\hat{\mathbf{T}} = \mathbf{T}_p\mathbf{T}$. We perturb the ground truth $\mathbf{T}$ using uniformly sampled $\mathbf{T}_p$, modeled by a translation and angle-axis vector. We sample these vectors uniformly on the 3-sphere using [20] and scale them with a uniformly distributed scalar. For training, the translation and rotation magnitudes are in the range $[0, 1.5]$m and $[0, 15]°$, respectively.

To account for different intensity profiles produced by different LiDAR models, and to cater for the fact that some LiDAR drivers don't provide intensity readings, we augment the intensity channel of our LiDAR data to minimize our dependence on intensity information. We apply uniform random scalar perturbations in the range $[0, 1.0]$ to the intensity channel of the LiDAR points, meaning that in some samples the intensity information is close to dropped out.

Lastly, to process the LiDAR data using our sparse 3-D CNN, we voxelize the points using isotropic voxels of 2 cm per side. We found this resolution to be reasonable since it is in the range of the measurement error reported by most automotive LiDAR manufacturers.

**Setup for camera input:** We've experimentally found that the camera feature extractor fails to learn generalizable geometric features unless spatial augmentations are applied. In our training, we performed random crop augmentations of [512, 256] pixels in width and height to the input image, and updated the camera intrinsic parameters accordingly.

## A.2   Model setup

Both image and point cloud feature extractors in our model follow the U-Net [14] architecture with 5 layers of coarse-to-fine features, each layer being a factor of 2 finer than the previous layer. We use features from 3 layers for our alignment, the 1/16-scale, the 1/4-scale, and the 1-scale. The dimensionality of the feature at these layers are 128, 128, and 32, respectively. To aid adaptation, at each pyramid level, the features from both domains are passed through a shared 2-layer MLP with the same input and output dimensions at each layer and Leaky ReLu activation.

With recent advancements in 3-D convolutions [21], we've chosen the spconv [22] implementation of sparse 3-D CNNs for our LiDAR feature extractor, as it can efficiently handle large point clouds, and can effectively manage the sparsity pattern of the data.

Learning features in the image domain is relatively straightforward, we've found that using the same U-Net [14] architecture (similar to Pixloc [11]) with the 2-D convolutional VGG [23] backbone was sufficient for image feature learning.

We initialize the visual extractor using weights from a pre-trained PixLoc [11] model trained on the CMU Seasons dataset [24], and the LiDAR extractor using only the sparse 3D CNN weights of a SphereFormer [25] model pre-trained on Semantic KITTI [26]. For the pose optimization, we trained with $M = 5$ iterations at every pyramid level.

# B   Additional Evaluations

## B.1   Calibration recall as a function of initial error

To gauge the robustness of our method to initial calibration errors, we conducted an experiment to measure the percentage of calibration trial that achieve errors less than 2cm and $0.1°$in both translation and rotation, respectively. In plotting this percentage as a function of the initial calibration error, we aim to show the sensitivity of our method to initial calibration errors both in the single sample optimization setting and in the batched optimization setting. We used the unseen camera from the held-out KITTI Odometry data sequence. The results plotted in Figure 5, show that batched optimization significantly boosts the robustness when there are large initial errors, achieving accurate calibration over 60 percent of the time even when the initial error is in the $±2m, ±20°$range unseen during training.

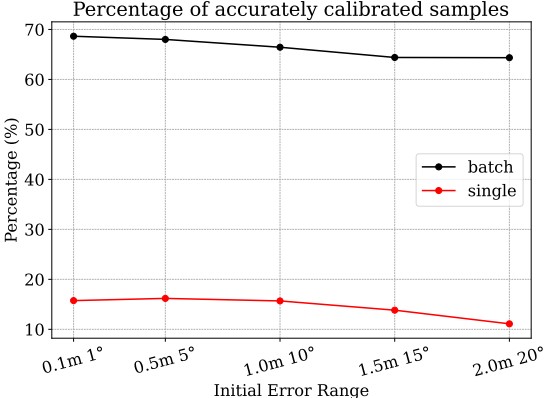

Figure 5: A plot of the percentage of calibration results that have errors less than 2cm and $0.1°$, as a function of the initial error range, tested on the unseen camera from the held-out sequences of the KITTI Odometry dataset. As seen, the batched optimization (batch size of 8) can estimate an accurate calibration over 60 percent of the time, even when tested in the initial error range of $±2m, ±20°$which was not encountered during training.

## B.2   Comparison against additional methods

We compared our method to LCCNet and DXQNet in our evaluations as they have the best performance for two different key traits — LCCNet excels in recovering from large initial errors, and can DXQNet transfer to unseen environments. Of our cited works, there is only one other method based on differentiable alignment, RGKCNet, but this method has reported lower calibration accuracy than DXQNet, in a slightly different experiment setting. For completeness, we also show the performance of our method tested in the RGKCNet experiment setting on the KITTI Odometry dataset. The results, summarized in Table 4 show that our method significantly outperforms RGKCNet in both translation and rotation metrics. Note that the median metrics for CalibNet are not included because they were not made available by the authors.

Table 4: Comparison against RGKCNet and CalibNet in the KITTI Odometry setting. Initial error in the range: $±7.5°±0.2m$

| Method | Mean/Median $\Delta t$ (cm) | | | Mean/Median $\Delta R$ (°) | | |
|---|---|---|---|---|---|---|
| | x | y | z | roll | pitch | yaw |
| CalibNet | 12.0/_ | 3.5/_ | 7.9/_ | 0.18/_ | 0.9/_ | 0.15/_ |
| RGKCNet | 5.0/2.8 | 4.0/2.6 | 5.9/3.4 | 0.16/0.09 | 0.15/0.10 | 0.17/0.11 |
| Ours (1) | 3.2/1.0 | 2.7/1.0 | 3.4/1.2 | 0.09/0.05 | 0.13/0.05 | 0.14/0.04 |
| Ours (8) | **0.4/0.3** | **0.8/0.8** | **0.6/0.5** | **0.02/0.02** | **0.04/0.04** | **0.02/0.02** |

## B.3    The impact of batch optimization

### B.3.1    A closer look at results from Table 2

In the case of using only a single image and point cloud pair, the direct alignment can encounter outlier cases where the optimization diverges. In the absence of other sources of error, the median metric can be robust to these outlier cases. However, in addition to spurious diverging outlier cases, direct alignment with batch size = 1 is also affected by the frequent convergence to local optima. In autonomous driving scenarios, these local optima are prevalent along the translation axes. Due to the lack of features close to the camera, changes to the translation parameters create minimal visual parallax which can make the optimization less sensitive to translation errors.

This shortcoming of direct alignment is consistent with our findings reported in Table 2. Referring to the median rotation metric on the unseen camera, we see that Ours (1) performs just as well as DXQNet, and nearly an order of magnitude better than LCCNet (the regression-based method). On the other hand, on the median translation metric for the unseen camera, Ours (1) performs an order of magnitude better than the regression-based method LCCNet (as we expected), but is still 6 millimeters less accurate than DXQNet (the sparse flow-based approach).

In this context, optimizing over a batch with size greater than 1, not only decreases the impact of outlier diverging samples but also improves the accuracy of all samples altogether. This can be seen in the histograms of the error distributions for translation and rotation in Figure 6. The error distribution of the batch optimization (size 8) exhibits a shorter tail, it also has a sharper peak closer to zero.

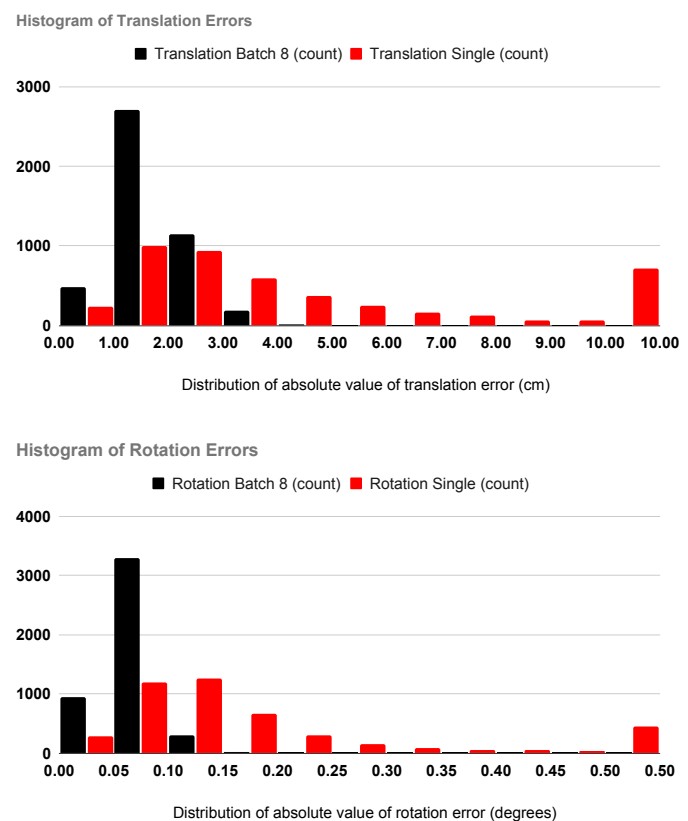

Figure 6: The error histograms of our method on the unseen camera referred to in Table 2. In both translation and rotation error, the batched approach exhibits a shorter tail and also peaks closer to zero than the single sample optimization. The last red bin in each plot aggregates all larger (outlier) values.

### B.3.2 Experiment with 3 simple scenes

To demonstrate how batch optimization can improve accuracy, even on non-diverging samples, we performed a small but instructive experiment using 3 image/point-cloud pairs from the validation set of KITTI odometry. Each of these samples contains sufficient information such that direct alignment does not diverge to extreme errors. To further avoid divergence, we sampled initial guess transforms relatively close to the ground truth (within 50 cm and 5 degrees).

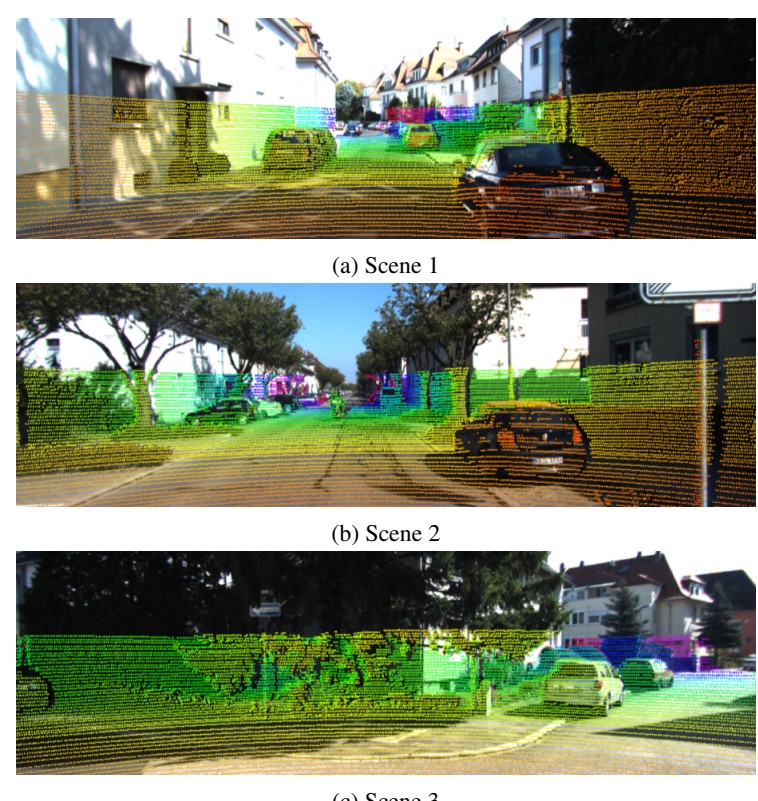

(a) Scene 1

(b) Scene 2

(c) Scene 3

Figure 7: Scenes used in experiment B.3.2: Scene 1 has relatively more features close to the camera, whereas Scene 3 has the fewest features due to a large patch of under-exposed bush in the image.

Running individual optimizations from 100 different initializations yields the distribution of translation errors seen in Figure 8.

Of the three scenes, performance is best on Scene 1 as it has more features closer to the camera whereas performance is poorest in Scene 3 as it has the sparsest features. Aggregated statistics are shown in Table 5. The last column shows the mean and median values found by stacking all error values from scenes 1, 2, and 3 together. Note that the values in the last column are close to the values computed from Scene 2, the mid ranking scene.

Table 5: Aggregate error statistics over 100 different runs of individual optimization on each of the 3 scenes from Figure 7

| Translation Errors on the Experiment with 3 Scenes (cm) | | | | |
|---|---|---|---|---|
| | Scene 1 | Scene 2 | Scene 3 | Scenes Stacked |
| Mean | **1.49** | 3.10 | 6.52 | 3.72 |
| Median | **1.11** | 3.05 | 6.36 | 2.96 |

Performing the same experiment as above, only running batch optimization with all 3 scenes yields significantly better results, as seen in Figure 9.

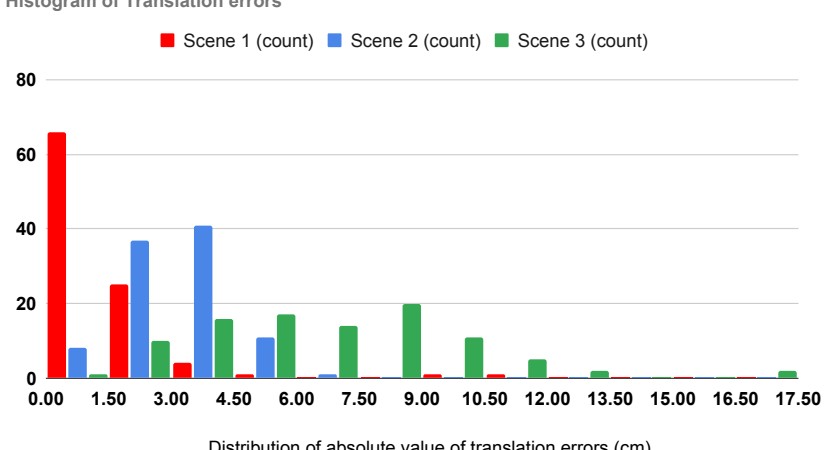

Figure 8: Error distribution of optimizing 100 different initializations of each of the scenes in Figure 7 individually.

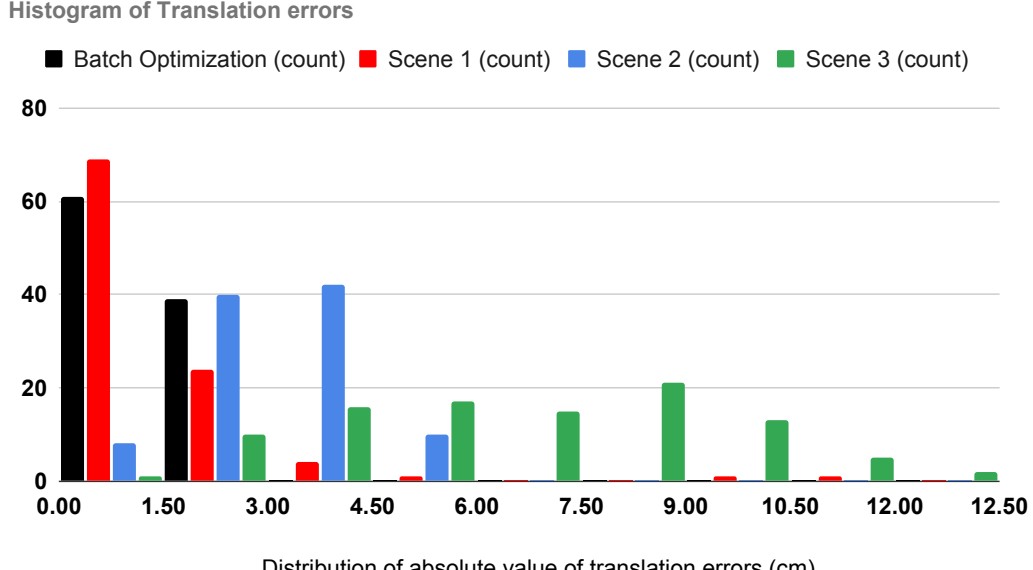

Figure 9: Error distribution of optimizing 100 different initializations of all scenes in Figure 7 jointly (Batch Optimization) compared against the individual optimizations. The distribution of the batch optimization errors exhibits a shorter tail and is closer to zero.

By fusing information from all scenes, the overall error distribution is sharper and closer to zero than the error distribution of any of the individual scenes. Table 6 shows the aggregate statistics found by performing batch optimization and compares it to the individual optimization result shown previously.

The last two columns of Table 6 show a very significant difference in mean and median error between performing batch optimization and individual per scene optimization. The mean error with batch optimization is better than the mean error achieved by any of the individual scenes. While the

Table 6: Aggregate error statistics over 100 different runs of batch optimization of all 3 scenes from Figure 7 jointly (Scenes Batch Optimized), contrasted against individual optimization.

| | Scene 1 | Scene 2 | Scene 3 | Scenes Stacked | Scenes Batch Optimized |
|---|---|---|---|---|---|
| Translation Errors on the Experiment with 3 Scenes (cm) | | | | | |
| Mean (cm) | 1.49 | 3.10 | 6.52 | 3.72 | **1.38** |
| Median (cm) | **1.11** | 3.05 | 6.36 | 2.96 | 1.28 |

median error of batch optimization is slightly worse than that of the best-performing individual scene, it is still significantly better than the median error of any other individual scene.

To conclude, the reason that batch optimization so significantly improves performance is not due to the rejection of outlier cases alone. Owing to the effective fusion of features in direct alignment, batch optimization yields solutions that are close in accuracy to the best individual sample in the batch. This effectively raises the performance of many samples, not just the outlier cases.

