# OpenReview forum: "Batch Differentiable Pose Refinement for In-The-Wild Camera/LiDAR Extrinsic Calibration"
_robot-learning.org/CoRL/2023/Conference — CoRL 2023 Poster_

### Official Review · Reviewer_sEmr · 2023-07-18

**Confidence:** 4
**Originality:** Good
**Technical Quality:** Very Good
**Clarity Of Presentation:** Excellent
**Impact:** 3

**Recommendation:**

Weak Accept: I recommend accepting the paper, but will not argue for my recommendation if the majority of other reviewers have a different opinion.

**Review:**

**Strengths :**

* The method shows a very good level of technical quality. Direct alignment on deep features has shown success in other areas such as camera relocalization and applying it to camera-lidar calibration is relevant and has not been done before to the best of my knowledge. Performing this alignment in a batched setup is a simple idea but well executed (using the same initial offset for different sensor setups during training is smart) and clearly beneficial.


* The results with batched optimization are comparable or better than competitors and exhibit good generalization properties. The competitors are also well selected.


* The paper is well writen and presented.

**Weaknesses :**

* I am really surprised by the difference between the error with and without batch optimization. It is intuitive that using more data brings more information and improves the optimization process. However, the results with a single pair of sensor data are really lower than expected. Competitors use only one pair of image/scan but exhibit better results. Why is the difference between "Ours (1)" and "Ours (8)" that large in all tables ? For example I would expect direct alignment to perform way better than the regression-based competitor on a new camera (Table 2).

* As accurately pointed out in the limitations, the method assumes exact time synchronization between the camera image and lidar scan, which is usually the case in public dataset but much more difficult in practice (or it could be considered part of the calibration process). Recent work on the same topic [1] is also able to estimate the time offset between the sensors, is it something that could be improved in the future with this method ?

[1] "MOISST: Multi-modal Optimization of Implicit Scene for SpatioTemporal calibration" Herau et al. arxiv 2023

**Quality Of The Limitations Section:**

Limitations are addressed clearly

**Questions For Rebuttal:**

I would really like more explanations and clarifications on the first weakness pointed out above. Am I missing something here ?
 I know that direct alignment can fail to converge when priors are too far from the solution, leading to failure cases and degrading the mean error, but even the median error is not very good for the optimization with batch size = 1. I find it suprising that increasing the batch size can improve the results that much.
Addressing this weakness in the rebuttal is way more important in my opinion than emphasize on the second weakness, which is already mentioned in the limitations section and more a remark from my side than something that prevents the paper to be published.

**Robotics Focus:**

Highly relevant to robotics but no hardware experiments

**Summary Of Paper:**

The paper tackles target-free camera-lidar extrinsics calibration with a learning-based approach. The problem is solved by direct alignment between deep features coming from a 2D CNN for the image and a sparse 3D CNN for the lidar scan. A key component of the method is to optimize the relative pose between sensors in a batch of image/scans pairs. Experiments show more accurate results than relevant competitors for several experimental scenarios, evens with sensors unobserved during training.

**Summary Of Recommendation:**

Overall, I think that this is a paper a good quality thanks to a polished writing, a well executed technical solution with some novelty and it is clearly relevant to the robotics community. My current recommendation is weak reject because of the poor results of the non-batched version, which seem counter intuitive and need to be explained more clearly in order to understand why the batched method performs well and better than competitors. I am disposed to increase my score if this problem is addressed.

Post rebuttal : My concerns have been addressed thoroughly during the rebuttal and I am now convinced that the proposed method is worth publication.

---

> ### Author Response · Authors · 2023-08-13
> **Response to Reviewer sEmr**
>
> Thank you for your time and appreciation for our work!
>
> **Q1**: Why is the difference between "Ours (1)" and "Ours (8)" that large in all tables? For example I would expect direct alignment to perform way better than the regression-based competitor on a new camera (Table 2).
>
> **A1**: In the case of using only one image/point cloud pair, the direct alignment can encounter outlier cases where the optimization diverges. In the absence of other sources of error, the median metric can be robust to these outlier cases. However, in addition to spurious diverging outlier cases, direct alignment with batch size = 1 is also affected by the frequent convergence to local optima. In autonomous driving scenarios, these local optima are prevalent along the translation axes. Due to the lack of features close to the camera, changes to the translation parameters create minimal visual parallax – making the optimization less sensitive to translation errors.
>
> This shortcoming of direct alignment is consistent with our findings reported in Table 2.
> Referring to the median rotation metric on the unseen camera, we see that Ours (1) performs just as well as DXQNet, and nearly an order of magnitude better than LCCNet (the regression-based method). On the other hand, on the median translation metric on the unseen camera, Ours (1) performs (as expected) an order of magnitude better than the regression-based method LCCNet, but is just 6 millimeters less accurate than DXQNet (the sparse flow-based approach).
>
> In this context, optimizing over a batch of size greater than 1, not only decreases the impact of outlier diverging samples but also improves the accuracy of all samples altogether. We have added **Figure 6** in **Appendix B.3.1** (added to the updated manuscript), to demonstrate how the error distribution of the batched optimization not only exhibits a shorter tail but also a peak that is closer to zero --- showing that batched optimization is doing more than just preventing outlier samples from diverging.
>
> Furthermore, as we show in our new experiment in **Appendix B.3.2** (added to the updated manuscript), owing to the effective fusion of features in direct alignment, the batched optimization yields solutions that are close in accuracy to the best individual sample in the batch. This effectively lifts the accuracy of many samples, not just the outlier cases.
>
> **Q2**: As accurately pointed out in the limitations, the method assumes exact time synchronization between the camera image and lidar scan, which is usually the case in public dataset but much more difficult in practice (or it could be considered part of the calibration process). Recent work on the same topic [1] is also able to estimate the time offset between the sensors, is it something that could be improved in the future with this method ?
>
> **A2**: Yes, this fits well into our plan of incorporating additional consistency priors into our feature learning process. The differentiable representation of per-sensor relative pose from [1] will be helpful in our future work on joint ego-motion/temporal/multi-sensor calibration. We have updated this observation in our limitations section in the updated manuscript.
>
> [1] "MOISST: Multi-modal Optimization of Implicit Scene for SpatioTemporal calibration" Herau et al. arxiv 2023

---

> > ### Comment · Reviewer_sEmr · 2023-08-14
> > **Aknowledgment of rebuttal**
> >
> > I acknowledge the rebuttal from the authors and appreciate the detailed answers to my comments.
> > In particular, I like the experiment in Appendix B.3.2 that supports the explanation provided in the rebuttal and provide interesting insights on the method. I will analyze it further and will consider to increase my score during the reviewer discussion period.

---

### Official Review · Reviewer_3776 · 2023-07-19

**Confidence:** 4
**Originality:** Very Good
**Technical Quality:** Very Good
**Clarity Of Presentation:** Very Good
**Impact:** 3

**Recommendation:**

Weak Accept: I recommend accepting the paper, but will not argue for my recommendation if the majority of other reviewers have a different opinion.

**Review:**

- Originality: the paper is well motivated, pin-pointing to the unique novelty of the proposed approach with respect to existing literature. Related work is widely surveyed and well cited.
- Quality: the proposed framework is technically sound, backed up by comprehensive evaluations. The zero-shot transfer results are impressive.
- Clarity: the paper is well written, easy to follow.
- Significance: automated lidar-camera calibration is an important problem. The paper has a potential of real-world deployment.
- Relevance: the online mapping problem is critical in the automotive industry. The paper is definitely on the right path.
- Limitations: Discussion section clearly states the assumption of shared visibility, the limitation of not leveraging ego motion, and lower accuracy compared to traditional offline methods.


**Quality Of The Limitations Section:**

Limitations are addressed clearly

**Questions For Rebuttal:**

- While the authors have compared the proposed approach with two existing work, separately, with different perturbation parameters, it would be nice to make them consistent unless there is a convincing reason not to do so, to avoid the doubt of cherry picking results.
- It would be nice if more baselines can be included beyond DXQNet, and LLCNet. Many cited in related work, but only two evaluated and compared against.
- If space permitted, touching a bit on the requirement/dependency of good initial guess would also be greatly appreciated. Knowing how much of  a bad initial guess can be tolerated, and how faster the iteration can converge to good performance is valuable to real-world deployment,


**Robotics Focus:**

Highly relevant to robotics but no hardware experiments

**Summary Of Paper:**

This paper proposed a batch differentiable pose refinement network for camera lidar calibration. The proposed approach targets targetless calibration, argues for decoupling feature extraction and calibration to avoid scene memorization and overfitting, and therefore highlights its capability of zero-shot transfer to unknown camera, lidar and environment. The approach is an iterative one, refining previous/initial transformation, and determines the loss based on the Jacobian of transformed pixel location difference with respect to the transformation parameters themselves. Evaluations show batch version outperforms state-of-the-art approaches, and it is capable of zero-shot transfer to different sensors and datasets.

**Summary Of Recommendation:**

This work proposed a batch differential pose refinement approach, advancing the state-of-the-art on targetless camera-lidar calibration. Evaluation is reasonably convincing and zero-shot transferability is a highlight of the paper.

---

> ### Author Response · Authors · 2023-08-13
> **Response to Reviewer 3776**
>
> We are thankful for the kind comments and appreciation for our work.
>
> **Q1**: While the authors have compared the proposed approach with two existing work, separately, with different perturbation parameters, it would be nice to make them consistent unless there is a convincing reason not to do so, to avoid the doubt of cherry picking results.
>
> **A1**: We are grateful for this feedback and have updated **Table 1 and Table 2** to show evaluations of LCCNet in all KITTI odometry experiment settings. We have not included DXQNet in the harder experiment settings with larger initial error ranges because the authors of DXQNet declared that the model is only meant for calibration recovery from small drifts ±0.1m\±5◦. In contrast to DXQNet, our model not only transfers, zero-shot, to unseen sensors/environments but also recovers calibration from large initial offsets e.g. the ±1.5m\±20◦ range shown in Tables 1 and 2.
>
> **Q2**: It would be nice if more baselines can be included beyond DXQNet, and LLCNet. Many cited in related work, but only two evaluated and compared against.
>
> **A2**: We initially included only LCCNet and DXQNet as they have the best
> performance for two different key traits – LCCNet excels in recovering from large initial errors, and DXQNet transfers to unseen environments. Of our cited works, there is only one other method based on differentiable alignment, RGKCNet, but this method has reported lower calibration accuracy than DXQNet, in a slightly different experiment setting. In our revision attached to this rebuttal, we have evaluated our method using the experimental setup reported in RGKCNet and have included the results in **Appendix B.2** where our method demonstrates significantly better performance in both rotation and translation.
>
> **Q3**: If space permitted, touching a bit on the requirement/dependency of good initial guess would also be greatly appreciated. Knowing how much of a bad initial guess can be tolerated, and how faster the iteration can converge to good performance is valuable to real-world deployment,
>
> **A3**: Indeed, we agreed that it would be beneficial to get an idea of the robustness of our method to the initial error range. To this end, we have performed a new experiment showing the percentage of accurate calibration as a function of the initial error range which has been added to **Appendix B.1**. Here accurately calibrated samples are those whose translation and rotation errors are each lower than 2cm and 0.1◦, respectively. We see that our method
> retains accurate calibration over 60% of the time even from an initial error of ±2m\±20◦.

---

> > ### Comment · Reviewer_3776 · 2023-08-14
> > **Acknowledging the rebuttal**
> >
> > The authors have addressed the comments well and revised the submission accordingly.

---

### Official Review · Reviewer_StF3 · 2023-07-20

**Confidence:** 3
**Originality:** Very Good
**Technical Quality:** Very Good
**Clarity Of Presentation:** Good
**Impact:** 3

**Recommendation:**

Weak Accept: I recommend accepting the paper, but will not argue for my recommendation if the majority of other reviewers have a different opinion.

**Review:**

I'm not very familiar with this topic, espcially in the 2 baselines (LCCNet, DXQNet),  but I can understand the approach overall.
## Strength
1. An interesting and important task.
2. The ability to restore the calibration error from >1m in input to a few cm is very impressive. Especially given that it can be applied on novel sensor placements and new dataset

## Weakness
1. The writing is not that clear to me (maybe because I'm not very familiar with this). It's not fluent either, e.g. the authors mix past tense and present tense here and there.
2. The improvement seems a bit incremental compared to DXQNet,  The improvement of accuracy is  a few millimeter, and 0.2 degree -- seems negligible.  And this improvement is at the cost of using batched optimization.

**Quality Of The Limitations Section:**

Limitations are addressed clearly

**Questions For Rebuttal:**

Please address my concern on the weakness part. Besides:

1. Why in setting ±1.5m/±20◦, only baseline LCCNet is applied and in ±0.5m\±10◦, only baseline DXQNet is applied?
2. What's the difference  between Mean ∆t in table 2 and  mean ∆t in table 1? Why the number is significant different. For example, 0.24 Vs 1.59 for LCCNet.

**Robotics Focus:**

Relevant but unlikely to deploy to hardware in near future

**Summary Of Paper:**

This work is about pose-refinement for self-driving scenarios, in special, the authors focus on zero-shot transfer of pose-refinement model: trained model can be applied on new scenes, new sensor placements and configurations.
The basic ideas are to align deep features between lidar and images.
Experiments are conducted on KITTI, KITTI360 and Waymo to deonstrate the effectiveness of the method.


**Summary Of Recommendation:**

Overall I think the method makes sense, the results are good. But I think the improvement is a bit incremental compared to DXQNet.

---

> ### Author Response · Authors · 2023-08-13
> **Response to Reviewer StF3**
>
> We are grateful for the encouraging comments on our work!
>
>
> **Q1**: The writing is not that clear to me (maybe because I'm not very familiar with this). It's not fluent either, e.g. the authors mix past tense and present tense here and there.
>
> **A1**: Following the feedback regarding the clarity of writing, we have done our best to rid our manuscript of any mix of tenses. These changes have been incorporated in the updated manuscript.
>
> **Q2**: The improvement seems a bit incremental compared to DXQNet, The improvement of accuracy is a few millimeter, and 0.2 degree -- seems negligible. And this improvement is at the cost of using batched optimization.
>
> **A2**: Compared to DXQNet, our main advances are in the transferability to other datasets and the ability to recover from larger initial errors. When transferred to the KITTI-360 dataset, our batched method performs significantly better than DXQNet achieving a 1cm improvement in median translation error and 0.5◦ improvement in median rotation error.
>
> While many methods prior to DXQNet (e.g. LCCNet) have all been evaluated in the large initial error range of ±1.5m\±20◦, DXQNet has by design only been evaluated in the small initial error region of ±0.1m\±5◦. Our work is an advancement over DXQNet in that we do demonstrate competitive calibration accuracy in the large initial error settings, while also transferring zero-shot to unseen environments.
>
> **Q3**: Why in setting ±1.5m/±20◦, only baseline LCCNet is applied and in ±0.5m\±10◦, only baseline DXQNet is applied?
>
> **A3**: We originally compared against LCCNet and DXQNet separately because these two methods individually exhibit different strengths – LCCNet can recover calibration from a large initial error but does not transfer to unseen sensors, whereas DXQNet does transfer to new sensors/environments but only operates within a very narrow initial error range. In comparing against LCCNet and DXQNet separately, we wanted to show that our method has both of these key traits: recovering from large initial errors and transferring to new environments.
>
> According to the authors of DXQNet, the network is designed to only correct for minor online calibration drifts in the narrow range of ±0.1m\±5◦, which is a significantly narrower range compared to the ±1.5m\±20◦ setting demonstrated by LCCNet (as well as many prior works). DXQNet has not been open-sourced for evaluation on new datasets, so the metrics we cited come directly from the DXQNet manuscript, where the authors demonstrated performance in the narrow initial error range of  ±0.1m\±5◦.
>
> Due to an error on our part, we previously misquoted DXQNet’s initial calibration error range as a wider range – meaning that we were evaluating our method in a harder setting than DXQNet’s experiment setting. We have rectified this error and have updated **Tables 1 and 2** with the results of running our method using the correct DXQNet evaluation setting. This narrower range is a simpler calibration problem than we previously evaluated, thus none of our observations and conclusions about our method have changed. We regret making this mistake in our initial submission. After thorough verification, we are confident in the results of the updated manuscript.
>
> Following the feedback, we’ve added the evaluation of LCCNet in the narrower initial error setting, using the official implementation of LCCNet (https://github.com/IIPCVLAB/LCCNet). As seen in the updated Table 2, even in this easier initial error setting, LCCNet performs well on the seen camera but its performance drops significantly when transferred to the unseen camera.
>
> **Q4**: What's the difference between Mean ∆t in table 2 and mean ∆t in table 1? Why the number is significant different. For example, 0.24 Vs 1.59 for LCCNet.
>
> **A4**: Table 1 and Table 2 differ in that Table 1 shows the per component error metrics e.g. x,y,z for translation, whereas Table 2 shows the metrics for the magnitude of translation and rotation errors. In the given example, the 0.24cm from Table 1 (mean of the x-component in translation error) is lower than the 1.59cm from Table 2 since Table 2 shows the mean of the magnitudes of the translation error vectors, which additionally includes the y and z components.

---

### Decision · Program_Chairs · 2023-08-30

**Decision:**

Accept (Poster)

**Comment:**

This paper presents a pose refinement network for camera/lidar extrinsic calibration. The method considers the targetless scenario and aims at zero-shot transfer for sensors and environments.

Strengths:
All reviewers agree that the problem is novel, sound, and relevant to the community.

Weaknesses:
Minor issues in the evaluation and discussion have been pointed out by the reviewers:
1) Minor improvement with DXQNet
2) Consistency in the perturbation parameters for the different benchmarks
3) Extra baselines
4) Time synchronization

----- Post rebuttal -----

In the rebuttal period, all reviewers agree that the authors have addressed most of their comments with the new sets of experiments, minor corrections to the language, and discussion in the limitations section.